# Association of Blood MCP-1 Levels with Risk of Obstructive Sleep Apnea: A Systematic Review and Meta-Analysis

**DOI:** 10.3390/medicina58091266

**Published:** 2022-09-13

**Authors:** Mohammad Moslem Imani, Masoud Sadeghi, Mehdi Mohammadi, Annette Beatrix Brühl, Dena Sadeghi-Bahmani, Serge Brand

**Affiliations:** 1Department of Orthodontics, Kermanshah University of Medical Sciences, Kermanshah 6715847141, Iran; 2Department of Biology, Science and Research Branch, Islamic Azad University, Tehran 1477893855, Iran; 3Students Research Committee, Kermanshah University of Medical Sciences, Kermanshah 6715847141, Iran; 4Center for Affective, Stress and Sleep Disorders (ZASS), Psychiatric University Hospital Basel, 4002 Basel, Switzerland; 5Department of Psychology, Stanford University, Stanford, CA 94305, USA; 6Sleep Disorders Research Center, Kermanshah University of Medical Sciences, Kermanshah 6715847141, Iran; 7Substance Abuse Prevention Research Center, Kermanshah University of Medical Sciences, Kermanshah 6719851115, Iran; 8Department of Sport, Exercise and Health, Division of Sport Science and Psychosocial Health, University of Basel, 4052 Basel, Switzerland; 9School of Medicine, Tehran University of Medical Sciences, Tehran 1417466191, Iran

**Keywords:** obstructive sleep apnea, blood, cytokine, monocyte chemoattractant protein-1, meta-analysis

## Abstract

Background and objective: Among the broad variety of chemokines, monocyte chemoattractant protein-1 (MCP-1) is considered to be one of the most important chemokines. Among others, MCP-1 activates monocytes and other immune cells highly involved in inflammation. In the present systematic review and meta-analysis, we evaluated the relationship between serum/plasma MCP-1 levels and the risk of obstructive sleep apnea (OSA) in adults as a disease related to inflammation. Materials and methods: Four databases were systematically investigated until 12 July 2022. We used the Review Manager 5.3 software (Copenhagen: The Nordic Cochrane Centre, The Cochrane Collaboration, Copenhagen, Denmark) to extract and calculate the standardized mean difference (SMD) and its 95% confidence interval (CI) of plasma/serum levels of MCP-1 between adults with and without OSA. Results: Eight articles including eleven studies in adults were entered into the meta-analysis. The serum/plasma MCP-1 levels in adults with OSA were higher than that in the controls (SMD = 0.81; *p* = 0.0007) and as well as for adults with severe OSA compared to those with mild and moderate OSA (SMD = 0.42; *p* < 0.0001). The subgroup analysis showed that ethnicity was an effective factor in the pooled analysis of blood MCP-1 levels in adults with OSA compared to the controls (Asians: (*p* < 0.0001), mixed ethnicity: (*p* = 0.04), and Caucasians: (*p* = 0.89)). The meta-regression showed increasing serum/plasma MCP-1 levels in adults with OSA versus the controls, publication year, age of controls, body mass index (BMI) of controls, and sample size reduced, and also BMI and the apnea–hypopnea index of adults with OSA increased. Conclusions: The meta-analysis showed that compared to the controls, serum/plasma levels of MCP-1 in adults with OSA were significantly more, as well as adults with severe OSA having more serum/plasma MCP-1 levels compared to the adults with mild to moderate OSA. Therefore, MCP-1 can be used as a diagnostic and therapeutic factor in adults with OSA.

## 1. Introduction

Approximately one billion adults aged between 30 and 69 years are suffering from obstructive sleep apnea (OSA) [1]. A prevalence rate of 9–38% has been shown for OSA, when measured with an apnea–hypopnea index (AHI) of ≥5 events/h [2], and from 6–17%, when measured with an AHI ≥15 events/h [2]. Male gender [2] and overweight are major risks for OSA [3,4,5].

Further, processes of inflammation occur as a consequence of exposure to tissues and organs against damaging stimuli, such as microbial pathogens, stimulants, or toxic cells [6]. The link between OSA and systemic inflammation is strong, and efforts are made to clarifying the causal correlations between OSA and inflammatory processes and to recognizing potential biological markers [7,8,9,10]. Chronic intermittent hypoxia and sleep fragmentation can increase the chronic inflammatory response; therefore, a chronically stimulated immune system can be substantial in the OSA pathogenesis [11].

Chemokines (chemotactic cytokines) are the only group of cytokines that interact with G protein-coupled receptors [12]. Chemokines are small binding proteins (60 to 100 amino acids) and structurally similar to cytokines [13]. Inflammatory chemokines regulate the deployment of effector leucocytes in tissue injury, infection, inflammation, and cancers [14]. Lots of these chemokines selectively affect the target cell and take action on the cells of the adaptive and innate immune system [14]. As such, chemokines are considered important agents in the process of inflammation and autoimmune response; indeed, chemokines appear to be highly involved in the lymphoid organogenesis, angiogenesis, and immune regulation [15]. In this vein, recent meta-analyses showed the association between several polymorphisms in adults with OSA in comparison to adults without OSA [16,17,18,19].

Among the broad variety of chemokines, monocyte chemoattractant protein-1 (MCP-1) is included in the family of CC chemokines [20]. An alternative expression to MCP-1 is Chemokine (CC-motif) ligand 2 (CCL2). MCP-1/CCL2 is considered one of the main chemokines responsible for the regulation of the migration and infiltration of monocytes/macrophages [13]. MCP-1 on binding to its receptor (C-C Motif Chemokine Receptor 2 (CCR2)) activates monocytes and immune cells highly involved in the inflammation response [13]. Studies have reported different results for the association between OSA risk and blood levels of MCP-1, specifically, while some studies showed high serum/plasma levels of MCP-1 in adults with OSA in comparison with controls [21,22,23], in contrast, other studies were unable to confirm such a pattern of results [24,25], or even reported opposite results [26]. Further, to our understanding, no thorough meta-analysis and systematic review were so far performed. With this context in mind, the aims of the present meta-analysis and systematic review were two-fold: First, to compare MCP-1 levels between adults with OSA and the healthy population; second, to associate MCP-1 levels with the severity of OSA.

## 2. Materials and Methods

### 2.1. Study Plan

To perform the meta-analysis, we rigorously followed PRISMA guidelines [27]. We identified the PECO with the question [28,29]: Are blood MCP-1 levels different in adults with OSA compared with controls? This resulted in Human cases with/ without OSA: P; OSA disease: E; adults with OSA compared with controls: C; and variations in the plasma/serum MCP-1 levels: O.

### 2.2. Recognizing of Articles

One author (M.S) comprehensively searched the databases of Web of Science, the Cochrane Library, PubMed/Medline, and Scopus up to 12 July 2022, with an age restriction (just individuals with age ≥18 years old were included) to retrieve the related articles. The strategy of the search was: (“obstructive sleep apnea-hypopnea syndrome” OR “OSAHS” OR “obstructive sleep apnea” or “sleep apnea” or “OSA” or “obstructive sleep apnea syndrome” or “OSAS”) and (“monocyte chemotactic protein-1” or “monocyte chemotactic protein 1” or “monocyte chemotaxis protein-1” or “monocyte chemotaxis protein 1 “or “monocyte chemoattractant protein-1” or “monocyte chemotactic protein 1” or “MCP-1”). Moreover, the references or citations of the reviews in relation to the topic were checked to make sure that no study was lost and afterwards, the titles/abstracts of these articles were determined by the same author (M.S); after that, those that fully met the criteria were downloaded. The second author (M.M.I) re-examined the previous process. A disagreement among the authors was obviated by another author (S.B).

### 2.3. Suitability Criteria

Inclusion criteria were: (1) study with any design including adults with OSA and controls aged ≥18 years and not receiving any treatment; (2) studies reporting plasma/serum MCP-1 levels in diagnostic and therapeutic methods OSA and controls; (3) polysomnography was used to diagnose OSA, defined as AHI ≥5 events per hour for an adult; and (4) adults with OSA and controls had no other systemic diseases. Exclusion criteria were: (1) reviews, book chapters, meta-analyses, or conference articles; (2) studies with insufficient data; (3) studies with a lack of control group or the control group had AHI ≥5 events/h; (4) studies including cases aged <18 years old participants; and (5) studies involving adults with OSA and with other systemic diseases (e.g., diabetes and cardiovascular diseases).

### 2.4. Data Collection

Two authors (M.S and M.M.I) individually extracted the data of the articles entered into the meta-analysis. These data were the country and ethnicity of individuals, MCP-1 sampling, the sample size or number of adults with OSA and controls, quality score, OSA type, and the plasma/serum MCP-1 levels in two groups, and body mass index (BMI), age, and AHI means of two groups.

### 2.5. Quality Assessment

The quality evaluation of the studies was completed by one author (M.S) using the Newcastle–Ottawa scale (NOS), pursuant to which the number nine was the highest score for each study, and a score of ≥7 was considered as a high-quality score.

### 2.6. Statistical Analysis

The Review Manager 5.3 software extracted and calculated the standardized mean difference (SMD) and its 95% confidence interval (CI) of the plasma/serum levels of MCP-1 among the adults with OSA and without OSA. The Z-test was utilized to examine the pooled SMD significance (*p*-value (2-sided) < 0.05 was treated as significant). A *P*_heterogeneity_ less than 0.1 (I^2^ > 50%) was set to identify significant heterogeneity and therefore a random-effects model [30]. Further, a fixed-effect model [31] was used to identify if the heterogeneity was insignificant.

The subgroup analysis and the random-effect meta-regression analysis were performed according to three and seven variables, respectively.

Next, both “one-study-removed” and “cumulative analysis” were used to examine the stability/consistency of the pooled SMD. In addition, a Egger’s test via funnel plots was applied to determine the degree of publication bias [32] and Begg’s test showed if there was a significant correlation among the ranks of the estimations of effect and the ranks of their variances [33]. The *p*-values of both of the tests were calculated and a *p*-value (2-sided) < 0.10 suggested the existence of publication bias. These analyses were carried out applying the Comprehensive Meta-Analysis version 2.0 (CMA 2.0) software (Biostat, Englewood, NJ, USA).

To address false positive or false negative results in meta-analyses [34], trial sequential analysis (TSA) was accomplished using TSA software (version 0.9.5.10 beta) (Copenhagen Trial Unit, Center for Clinical Intervention Research, Copenhagen, Denmark) [35]. The required information size (RIS) was computed with alpha and beta risks of 5% and 20%, respectively, and a two-sided border type. We estimated heterogeneity (D^2^) = 97% for the plasma/serum MCP-1 levels. The mean difference calculated on empirical assumptions was produced automatically by the software. When the Z-curve crossed the RIS line or the borderline or entered into the futility zone, the number of individuals was large enough and the conclusion was trustworthy or dependable and therefore, no more information will be needed.

The GetData Graph Digitizer 2.26.0.20 (GetData Pty Ltd., Kogarah, Australia) software was applied to extract data from the graphs in some specific studies. The authors checked the last analyses and any disagreements were removed following discussion.

## 3. Results

### 3.1. Study Selection

To search in the databases as Figure 1 shows, 113 records were identified. Removing the duplicates and irrelevant records, 17 full-text articles were appraised for suitability criteria. Nine articles were removed for various reasons (three articles reported gene expressions of MCP-1, two articles reported polymorphisms of MCP-1, one article had a case group with type 2 diabetes, one article had a group with coronary artery disease, one article included a control group with AHI >five events/h, and one article included children.). Finally, eight articles including eleven studies were imported into the meta-analysis.

### 3.2. Characteristics

Table 1 presents the characteristics of the articles incorporated in the meta-analysis [21,22,23,24,25,26,36,37]. The articles were published from 2003 to 2021. Six studies [21,22,23,25,36,37] included Asian participants, one [24] included Caucasians, and one [26] included participants of mixed ethnicity. Five articles [21,22,24,25,36] analyzed the data in serum samples, and three [23,26,37] in plasma samples. Six articles [21,22,24,25,36,37] had high-quality scores (score ≥ 7). The number of cases and controls, mean AHI, BMI, and age are shown in Table 1. Three articles [22,23,24] reported data from two independent studies. One study [26] had 18 missed cases, but the authors did not indicate the number of cases missed in each individual group; given this, we considered the initial number of participants entering each group.

### 3.3. Pooled Analysis (Case vs. Control)

As Figure 2 presents, pooled SMD of serum/plasma MCP-1 levels was 0.81 for adults with OSA compared to controls (95% CI: 0.34, 1.27; *p* = 0.0007; I^2^ = 91% (*P*_heterogeneity_ < 0.00001)). The results showed significantly higher blood MCP-1 levels in the adults with OSA compared to controls.

### 3.4. Pooled Analysis (Adults with Severe vs. Mild/Moderate OSA)

As Figure 3 shows, pooled SMD of serum/plasma MCP-1 levels was 0.42 for adults with severe OSA compared to mild/moderate OSA (95% CI: 0.21, 0.62; *p* < 0.0001; I^2^ = 28% (*P*_heterogeneity_ < 0.23)). The result reported significantly higher blood MCP-1 levels in adults with severe OSA compared to mild/moderate OSA.

### 3.5. Subgroup Analysis

The subgroup analyses of MCP-1 levels in adults with OSA in comparison with the controls based on ethnicity, sample size, and sampling, are represented in Table 2. The results recommended that only ethnicity was an effective factor for pooled analysis. The serum/plasma MCP-1 levels in Asians with OSA were significantly more than the controls (*p* < 0.0001), as well as for mixed ethnicity (*p* = 0.04), but not for Caucasians (*p* = 0.89).

### 3.6. Meta-Regression

Table 3 reports the results of serum/plasma MCP-1 levels in adults with OSA in comparison with controls. The publication year, age, BMI, AHI, and sample size were confounding factors for the pooled analysis: increasing serum/plasma MCP-1 levels in adults with OSA versus controls, publication year, age of controls, BMI of controls, and the sample size reduced, and also the BMI and AHI of adults with OSA increased.

### 3.7. Sensitivity Analysis

Both the “one-study-removed” and “cumulative analysis” reported the stability of the pooled analysis of serum/plasma MCP-1 levels. Removing the articles [23,26] with a quality score of <7, the pooled SMD was 0.67 (95% CI: 0.38, 0.95; *p* < 0.00001; I^2^ = 56%); thus, the pattern of the results remained unaltered, though heterogeneity decreased (from 91% to 56%).

### 3.8. Publication Bias

The values of *p*-values for both Begg’s and Egger’s tests were 0.39180 and 0.00722, respectively (Figure 4). The results of the tests showed a moderate publication bias.

### 3.9. Trial Sequential Analysis

Figure 5 shows the TSA of the serum/plasma MCP-1 levels in adults with OSA versus controls. The result showed that there were sufficient cases for this analysis.

## 4. Discussion

The main results of the meta-analysis were that the serum/plasma level of MCP-1 including sufficient individuals in adults with OSA was significantly more than the serum/plasma level of MCP-1 of adult individuals without OSA. In addition, the serum/plasma level of MCP-1 in adults with severe OSA was higher than the serum/plasma level of MCP-1 in adults with mild/moderate OSA. Increasing the serum/plasma MCP-1 levels in adults with OSA versus the controls, publication year, age of controls, BMI of controls, and the sample size decreased, and also the BMI and AHI of adults with OSA elevated.

Among the broad variety of CC chemokines, MCP-1 is considered one of the most substantial CC chemokines. More specifically, inflammatory and stromal cells express MCP-1, while proinflammatory stimuli regulate the chemotactic activity of MCP-1 [38,39]. Animal and in vitro models showed that intermittent hypoxia can induce MCP-1 synthesis and expression via the activation of the NF-κB signaling pathway [40,41]. In addition, early studies showed that hypoxia caused by OSA can enhance the circulating MCP-1 levels, while in contrast, useful treatment for OSA can decrease the MCP-1 expression [36,42].

One study reported that the plasma levels of A proliferation-inducing ligand (APRIL) were significantly related to plasma MCP-1 in adults with OSA [37]. Another study [43] found a correlation between adipose tissue blood flow in adults and OSA for the gene expression of MCP-1. Deterrence of MCP-1 has been illustrated to improve insulin resistance and diminish macrophage infiltration among the adipose tissue of obese mice [44]. Further, OSA is independently related to the emergence of insulin resistance [45,46]. MCP-1 is upregulated in human atherosclerotic plaques, which appears to indicate the MCP-1 role in the increase and progression of early atherosclerotic lesions [47,48,49,50]. Next, OSA is closely associated with both various cardiovascular diseases (CVDs) and with atherosclerosis [51,52,53]. The connection between OSA with atherosclerosis and insulin resistance, and also the relationship between MCP-1 levels and atherosclerosis and insulin resistance appears to show a correlation between MCP-1 levels and OSA risk; similarly, the present meta-analysis showed high serum/plasma levels of MCP-1 in adults with OSA compared to the controls. Therefore, it appears pivotal to assess the basic mechanisms of the OSA–atherosclerosis and OSA–insulin resistance interaction.

One study [22] described that by increasing blood levels of MCP-1, AHI scores in adults with OSA were reduced; in contrast, the present meta-analysis yielded a different result, in that elevating the serum/plasma levels of MCP-1, meant that the BMI [23] and AHI scores [23,36] elevated. Further, the MCP-1 concentrations increased with higher obesity [54,55] and the MCP-1 levels increased with lower age in healthy people [56]. However, another study [57] reported that the serum MCP-1 levels were unrelated to the age of individuals with atherosclerosis (a disease linked with OSA [58]). In the present meta-analysis, increasing the age of the controls, blood MCP-1 levels reduced, and there was a lack of a significant correlation between the age of adults with OSA and blood MCP-1 levels. Therefore, the role of the participants’ age should be considered in future studies with more sensitivity and the role of AHI with less sensitivity.

In spite of the novelties, the below limitations should be remarked upon: (1) There was a low number of cases in most of the studies; (2) There was a high heterogeneity in the initial pooled analysis; (3) The occurrence of several confounding factors; (4) Some of the studies reported the MCP-1 level on the graphs and we estimated their means and standard deviations with appropriate software; and (5) There was a publication bias between or across the studies. In contrast, the meta-analysis included sufficient cases with stable results.

## 5. Conclusions

The findings showed that serum/plasma levels of MCP-1 in adults with OSA were significantly more than those in controls, as well as in adults with severe OSA compared to mild/moderate OSA. Therefore, it appears that MCP-1 can have a practical, clinical, and diagnostic importance in adults with OSA. In addition, age, ethnicity, BMI, and AHI should be considered in the diagnostic and therapeutic approach to adults with OSA.

## Figures and Tables

**Figure 1 medicina-58-01266-f001:**
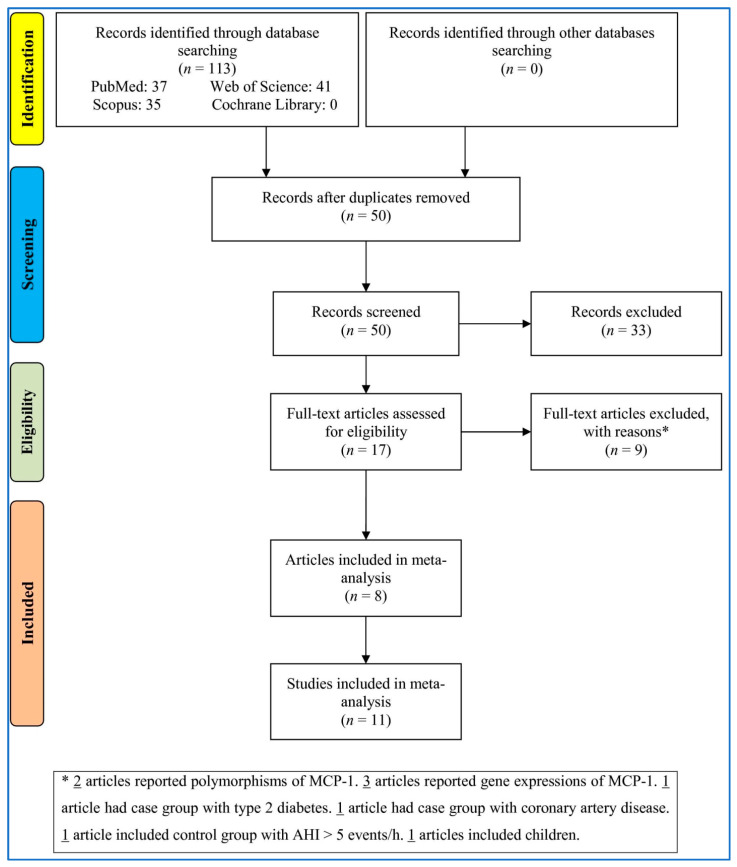
Flowchart of the study selection. MCP-1: Monocyte chemotactic protein-1. AHI: Apnea–hypopnea index.

**Figure 2 medicina-58-01266-f002:**
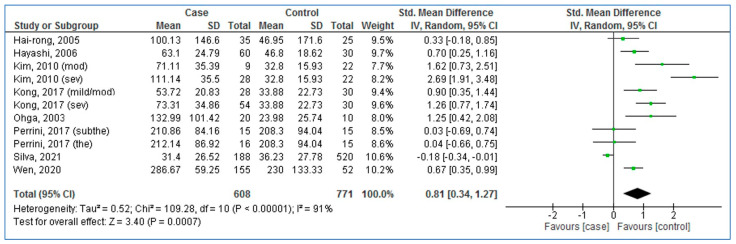
Forest plot analysis comparing blood levels of monocyte chemotactic protein-1 in adults with obstructive sleep apnea compared to controls. CI: Confidence interval. SD: Standard deviation. [21,22,23,24,25,26,36,37].

**Figure 3 medicina-58-01266-f003:**
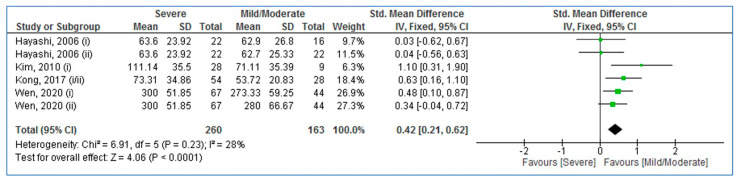
Forest plot analysis comparing blood levels of monocyte chemotactic protein-1 in mild/moderate compared to adults with severe obstructive sleep apnea. SD: Standard deviation. CI: Confidence interval [22,23,36,37].

**Figure 4 medicina-58-01266-f004:**
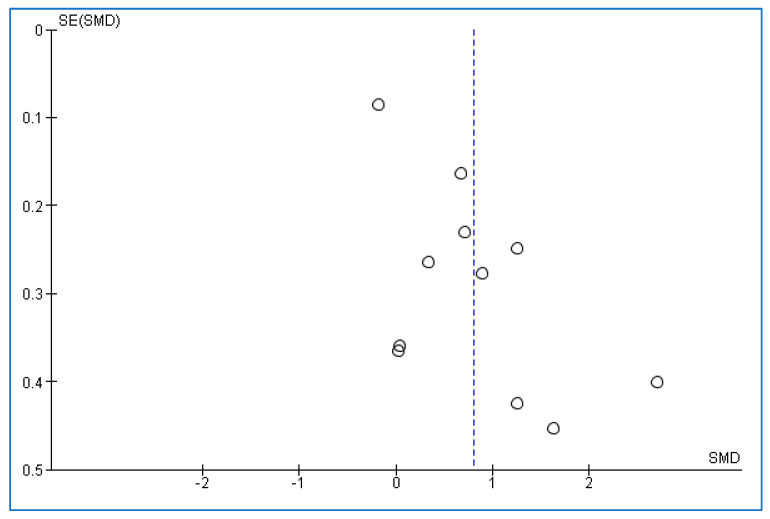
Funnel plot in adults with obstructive sleep apnea compared to controls. SMD: Standardized mean difference. SE: Standard error. Each small circle of the plot represents a separate one study. The vertical dotted line indicates the overall effect from the meta-analysis.

**Figure 5 medicina-58-01266-f005:**
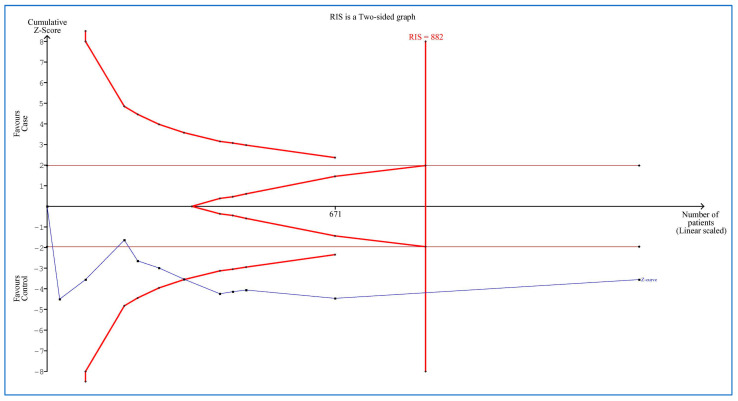
Trial sequential analysis in adults with obstructive sleep apnea compared to controls. The vertical red line shows the required information size (RIS). Horizontal brown lines shows conventional boundary for benefit (up) and for harm (down). Each black dot on the Z-curve (blue line) shows one study.

**Table 1 medicina-58-01266-t001:** Characteristics of articles.

First Author, Publication Year	Country	Ethnicity	Cases				Controls				Sampling	Quality Score
			No.	Mean			No.	Mean				
				AHI, events/h	BMI, kg/m^2^	Age, Years		AHI, events/h	BMI, kg/m^2^	Age, Years		
Ohga, 2003 [21]	Japan	Asian	20	38.9	29.4	47.8	10	3.1	28.4	48.9	Serum	9
Hai-rong, 2005 [25]	China	Asian	35	≥5	29.24	50	25	<5	28.04	50	Serum	8
Hayashi, 2006 [36]	Japan	Asian	60	49.4	28.8	51.6	30	2.8	23.2	55	Serum	7
Kim, 2010 * [23]	Korea	Asian	9	14.40	24.43	38	22	1.25	23.88	26	Serum	6
			28	52.71	28.69	42						
Kong, 2017 ** [22]	China	Asian	28	≥5	27.3	44.9	30	<5	27.6	45.7	Serum	9
			54	≥5	27.9	45.9						
Perrini, 2017 *** [24]	Italy	Caucasian	16	42.5	38	46.3	15	4.6	41.1	43.7	Serum	8
			15	27.3	38.7	46.1						
Wen, 2020 [37]	China	Asian	155	28.53	23.59	52	52	3.00	27.21	49.67	Plasma	7
Silva, 2021 [26]	Brazil	Mixed	188	26.97	25.1	47	520	4.8	28.7	45	Plasma	5

BMI, Body mass index; AHI, Apnea–hypopnea index. * This study included two studies (moderate vs. control and severe vs. control). ** This study included two studies (mild/moderate vs. control and severe vs. control). *** This study included baseline data in two studies (treatment cases vs. control and sub-treatment cases vs. control).

**Table 2 medicina-58-01266-t002:** Subgroup analysis in adults with obstructive sleep apnea compared to controls.

Variable, N	SMD	95% CI		*p*-Value	I^2^, %	*P* _heterogeneity_
		Min	Max			
Ethnicity						
Asian (8)	1.10	0.69	1.52	<0.00001	79	<0.0001
Caucasian (2)	0.03	*–*0.47	0.54	0.89	0	0.98
Mixed (1)	*–*0.18	*–*0.34	*–*0.01	0.04	-	-
Sample size						
<100 (9)	0.95	0.47	1.43	<0.0001	81	<0.00001
≥100 (2)	0.24	*–*0.59	1.07	0.58	95	<0.00001
Sampling						
Serum (9)	0.95	0.47	1.43	<0.0001	81	<0.00001
Plasma (4)	0.24	−0.59	1.07	0.58	95	<0.00001

N: Number of studies. SMD: Standardized mean difference. CI: Confidence interval.

**Table 3 medicina-58-01266-t003:** Meta-regression analysis in adults with obstructive sleep apnea compared to controls.

Variable	Point Estimate	Standard Error	Lower Limit	Upper Limit	Z-Value	*p*-Value
Publication year	−0.06450	0.01084	−0.08575	−0.04325	−5.94886	<0.00001
Mean age of adults with OSA	−0.03085	0.02243	−0.07481	0.01311	−1.37525	0.16905
Mean age of controls	−0.03391	0.01188	−0.05720	−0.01063	−2.85432	0.00431
Mean BMI of adults with OSA	0.02835	0.01804	−0.00701	0.06370	1.57128	0.00612
Mean BMI of controls	−0.07558	0.01744	−0.10976	−0.04139	−4.33352	0.00001
Mean AHI of adults with OSA	0.04198	0.00826	0.02579	0.05816	5.08275	<0.00001
Sample size	−0.00165	0.00020	−0.00204	−0.00126	−8.29125	<0.00001

## Data Availability

No new data were created or analyzed in this study. Data sharing is not applicable to this article.

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
