# Peer review of "Association of Blood MCP-1 Levels with Risk of Obstructive Sleep Apnea: A Systematic Review and Meta-Analysis"

_medicina, 2022, doi:10.3390/medicina58091266_

Round 1

Reviewer 1 Report

In general the manuscript "Association between serum and plasma MCP-1 levels and risk  of obstructive sleep apnea: A systematic review and meta anallysis"  is an manuscript which can be adaptted. 

I recommend a few modification.

Double check the English by a native English speaker.

Line146-160 ... it is not clear, I recommend to rephrase the entire paragraph.
Please describe the abbreviations when you use them for the first time

The figure  is extremely hard to read reformat in a clear format.

I  think that you need three short conclusions! My recommendation is to focus on short conclusion useful for clinicians.

Thank you again for the opportunity to review this interesting manuscript. 

Author Response

We thank Reviewer #1 for their encouraging and useful comments, which helped us to improve the quality of the manuscript. Please find attached the detailed point-by-point-response. Once again thank you very much for all your kind efforts.

Reviewer 2 Report

This is a systematic review and meta-analysis for assessing the relationship between serum and plasma levels of monocyte chemoattractant protein-1 and the risk and severity of obstructive sleep apnea in adults.

The manuscript is well written, has a good flow, and appropriate analysis was done.

I want to suggest a few modifications:

1. Line 54, “A chronic/oxygen intermittent sleeping and hypoxia fragmentation” is a bit confusing; I think “chronic intermittent hypoxia and sleep fragmentation” is more appropriate.

2. Line 98, citations

3. Line 113, please elaborate on other diseases such as xxx.

4. Line 252, something is missing at the end of the sentence.

Author Response

We thank Reviewer #2 for their encouraging and useful comments, which helped us to improve the quality of the manuscript. Please find attached the detailed point-by-point-response. Once again thank you very much for all your kind efforts.

Reviewer 3 Report

I have carefully read the literature review entitled "Association between serum and plasma MCP-1 levels and risk of obstructive sleep apnea: A systematic review and meta-analysis.

The authors have a good motodology in the search of the literature relevant to the topic of the review.

Adequate references were used and the authors were able to highlight all the salient points regarding the association between serum and plasma MCP-1 levels and risk of obstructive sleep apnea.

The work is well written and very clear.

The only important point is to improve native English.

Author Response

We thank Reviewer #3 for their encouraging and useful comments, which helped us to improve the quality of the manuscript. Please find attached the detailed point-by-point-response. Once again thank you very much for all your kind efforts.
